# Advancing Slim-Hole Drilling Accuracy: A C-I-WOA-CNN Approach for Temperature-Compensated Pressure Measurements

**DOI:** 10.3390/s24072162

**Published:** 2024-03-28

**Authors:** Fei Wang, Xing Zhang, Xintong Li, Guowang Gao

**Affiliations:** 1School of Electronic Engineering, Xi’an Shiyou University, Xi’an 710312, China; 21212030386@stumail.xsyu.edu.cn (X.Z.); wwgao@xsyu.edu.cn (G.G.); 2School of Mechanical and Electrical Engineering, Xi’an University of Architecture and Technology, Xi’an 710064, China; lxt146847@hotmail.com

**Keywords:** downhole pressure measurement, adaptive whale optimization, convolutional neural networks, temperature compensation

## Abstract

This paper presents a novel method to improve drill pressure measurement accuracy in slim-hole drilling within the petroleum industry, a sector often plagued by extreme conditions that compromise data integrity. We introduce a temperature compensation model based on a Chaotic-Initiated Adaptive Whale Optimization Algorithm (C-I-WOA) for optimizing Convolutional Neural Networks (CNNs), dubbed the C-I-WOA-CNN model. This approach enhances the Whale Optimization Algorithm (WOA) initialization through chaotic mapping, boosts the population diversity, and features an adaptive weight recalibration mechanism for an improved global search and local optimization. Our results reveal that the C-I-WOA-CNN model significantly outperforms traditional CNNs in its convergence speed, global searching, and local exploitation capabilities, reducing the average absolute percentage error in pressure parameter predictions from 1.9089% to 0.86504%, thereby providing a dependable solution for correcting temperature-induced measurement errors in downhole settings.

## 1. Introduction

In the evolving landscape of the petroleum industry, slim-hole drilling technology emerges as a pivotal advancement. It addresses the dual challenges of a rising hydrocarbon demand and the escalating costs associated with traditional drilling practices. Characterized by its reduced dependence on drilling consumables, this technology offers significant economic advantages in exploratory projects and minimizes environmental impacts. This has attracted considerable attention within the field. Moreover, in the drilling process, the monitoring of the drilling pressure is crucial for safety and efficiency, making the accurate measurement of the drilling pressure especially critical [1,2].

A critical aspect of drilling operations, especially under the severe high-temperature conditions typical of downhole environments, is the integrity of pressure measurements. Such extreme conditions often induce nonlinear deviations in pressure data, underscoring the necessity for advanced temperature compensation methods. Among the various solutions, software-based compensation stands out due to its cost-effectiveness, flexibility, and customization options, providing a practical alternative to the more cumbersome hardware-based approaches.

Despite their advantages, each software compensation method comes with inherent limitations. For instance, linear interpolation methods may introduce significant errors when applied beyond the original data range [3]. Polynomial fitting can effectively handle nonlinear sensor data but requires high-order polynomials, which are susceptible to overfitting [4]. Lookup table methods, reliant on extensive data and computational resources, often struggle with adaptability to new temperature conditions [5,6]. The BP neural network, while offering robust learning capabilities and an adaptability to sensor nonlinearities, may not perform well with complex datasets or in capturing spatial hierarchies [7,8]. CNNs excel in identifying local patterns and establishing spatial hierarchies, providing translational and spatial invariance, but can be hindered by slow convergence rates and a propensity for overfitting [9,10,11].

Addressing these challenges, this manuscript introduces the C-I-WOA-CNN model, an innovative integration of the Chaotic Whale Optimization Algorithm (CWOA) with the Integrated Whale Optimization Algorithm (IWOA), specifically tailored for the CNN framework. This hybrid approach leverages the strengths of the CWOA and IWOA, promoting faster convergence, enhancing global search capabilities, and facilitating more effective exploitation and an improved stability. The C-I-WOA-CNN model represents a sophisticated solution for temperature compensation in slim-hole drilling pressure measurement, offering significant advancements over existing methods [12,13].

## 2. Related Works

**Numerical analysis methods:** In the realm of numerical analysis, a plethora of techniques are employed to tackle various computational problems, among which Cubic Spline Interpolation, the Least Squares Method, and the Gradient Descent Method stand out for their effectiveness and widespread application. High-degree polynomial interpolation, while powerful, can sometimes lead to undesirable oscillations or inaccuracies, particularly in scenarios where data points are influenced by external factors, such as temperature-induced changes in drilling pressure. In such cases, piecewise interpolation methods, which segment the interpolation range into smaller intervals and apply lower-degree polynomials within each, prove to be more reliable. Unlike basic piecewise methods that may result in discontinuous or non-smooth junctions between segments, Cubic Spline Interpolation ensures a smooth transition by maintaining continuous second-order derivatives at the boundaries of each interval, thus providing a more accurate and visually pleasing interpolation [14,15].

The Least Squares Method, pivotal in the field of optimization, seeks to minimize the sum of squared residuals, representing the differences between observed values and those predicted by a model. This method is foundational in regression analysis and curve fitting, providing a robust approach to model estimation and prediction [16,17].

Gradient Descent, a cornerstone algorithm in both numerical analysis and machine learning, iteratively adjusts parameters in a model to minimize a defined loss function. This method is particularly effective for finding local minima in complex optimization problems, and its variants, such as Stochastic Gradient Descent and the Adam optimizer, have further enhanced its efficiency and applicability across various domains [18,19].

**Neural network methods:** The integration of optimization algorithms with neural networks has catalyzed significant breakthroughs in this field. Innovations such as Particle Swarm Optimization combined with Backpropagation Neural Networks (PSO-BP), Genetic Algorithms integrated with Convolutional Neural Networks (GA-CNN), and the fusion of Ant Colony Optimization with Backpropagation Neural Networks (ACO-BP) exemplify the synergy between optimization techniques and neural computation [20,21,22,23,24].

Particle Swarm Optimization, drawing inspiration from the social dynamics observed in bird flocks or fish schools, enhances the initialization and adjustment of neural network parameters, such as weights and biases, through a collaborative search mechanism. This approach not only improves the efficiency of the search process but also aids in circumventing suboptimal local minima, thereby enhancing the overall performance of BP neural networks [25,26].

Genetic Algorithms, modeled after evolutionary processes and natural selection, offer a powerful framework for optimizing the architecture and parameters of neural networks. By simulating the process of mutation, crossover, and selection among a population of potential solutions, GAs can effectively identify optimal or near-optimal configurations for CNNs, thus leveraging the evolutionary principles for computational advantage [27,28].

Ant Colony Optimization, inspired by the foraging behavior of ants, applies a probabilistic technique to explore and exploit solutions in the training of BP neural networks. By simulating the way ants deposit pheromones and follow the most promising paths to food sources, ACO algorithms diversify the search strategy of neural networks, enhancing their ability to escape local optima and converge towards globally optimal solutions [23,29].

## 3. Principle of Drilling Pressure Measurement

The methodology for drilling pressure measurement leverages the strain sensitivity of resistive strain gauges incorporated into a Wheatstone bridge circuit, a design chosen for its proficiency in translating minute changes in resistance into discernible electrical signals. In the operational framework, the imposition of drilling pressure induces a deformation in the strain gauge, manifesting as a variation in its electrical resistance. This alteration is pivotal, as the Wheatstone bridge, through its balanced circuitry, adeptly converts these resistance fluctuations into voltage outputs, thereby quantifying the drilling pressure with precision. Figure 1 expounds on the schematic representation of the Wheatstone bridge’s principle. It is imperative to note that the downhole environment, characterized by its elevated temperature conditions, can skew the strain gauge’s response, introducing nonlinear errors. To counteract this, a meticulous temperature compensation protocol is indispensable to ensure the fidelity and accuracy of the drilling pressure measurements [30,31].

## 4. Principle of C-I-WOA-CNN

### 4.1. Neural Network Methods

Addressing the challenge posed by temperature-induced nonlinearities in strain gauges, the application of Convolutional Neural Networks (CNNs) enables the modeling of intricate nonlinear relationships that are essential for precise temperature compensation [9]. This study proposes an optimized CNN architecture, comprising two convolutional layers, two pooling layers, ReLU activation functions, and a single fully connected layer, specifically tailored for drilling pressure measurement correction. Figure 2 conceptualizes the CNN framework tailored for temperature compensation, encapsulating an input layer, convolutional layers, activation layers, pooling layers, a fully connected layer, and an output layer. The CNN’s input layer accepts temperature data (Ut) from the drilling pressure measurement setup and the corresponding output voltage (UP), aiming to predict the drilling pressure (FP) based on the applied force (FN) to the strain gauge.

The convolutional layers, fundamental to the CNN, are tasked with the extraction of salient features from the input data. Considering the input’s one-dimensional nature against the desired singular output, a 1 × 3 convolutional kernel is employed for its efficacy in handling one-dimensional inputs while maintaining sensitivity to local data features. The ReLU activation function is selected for its proficiency in addressing the vanishing gradient problem that arises during the training of deep neural networks, as well as for its computational efficiency. It offers a straightforward yet potent nonlinear transformation that promotes quicker convergence and accelerates the training process. Moreover, ReLU’s capability to manage potential gradient explosion issues ensures the stability of the network architecture.

Pooling layers within the CNN play a crucial role in reducing the dimensionality of the feature maps, thereby enhancing feature extraction capabilities. In the context of temperature compensation, these layers aid the network in identifying temperature-invariant features, thus enhancing the model’s resilience to temperature fluctuations and reducing computational demands. The fully connected layer integrates the features derived from previous layers to execute the final regression or classification tasks. For temperature compensation, this layer empowers the CNN to generalize across varying temperature conditions, improving the robustness and adaptability of the model to different thermal environments. It enables the network to learn and adjust to temperature variations, facilitating effective temperature compensation across diverse operational settings [32,33,34].

### 4.2. Enhancing CNN with Whale Optimization Algorithm (WOA)

The Whale Optimization Algorithm (WOA) stands out as an innovative meta-heuristic optimization framework, drawing inspiration from the sophisticated hunting techniques of humpback whales. The appeal of meta-heuristic algorithms like the WOA lies in their gradient-free operation, which allows for a broad exploration of the solution space, thereby mitigating the risk of convergence to local minima—a common challenge in gradient-based optimization methods. These algorithms have found increasing application across various engineering domains due to their flexibility, simplicity in implementation, and robustness in navigating complex optimization landscapes. As depicted in Figure 3, meta-heuristic algorithms can be classified into three principal categories. The first category encompasses evolutionary-based intelligent optimization algorithms, such as Evolutionary Strategies (ESs) and Genetic Algorithms (GAs), which mimic natural evolutionary processes. The second category includes physics-inspired intelligent optimization strategies, exemplified by Simulated Annealing (SA) and the Black Hole Algorithm (BH), which draw analogies from physical phenomena to guide the search process. The third category comprises swarm intelligence-based optimization methods, such as Particle Swarm Optimization (PSO) and Ant Colony Optimization (ACO), which are inspired by the collective behavior of social organisms. Incorporating a Whale Optimization Algorithm (WOA) into CNN architectures presents a compelling strategy for refining network parameters. This enhancement strengthens the model’s ability to identify optimal feature representations, thereby boosting its overall efficacy in complex pattern recognition tasks, such as temperature compensation in drilling pressure measurements [35].

The Whale Optimization Algorithm (WOA) is a sophisticated optimization approach inspired by the social and hunting strategies of humpback whales, particularly their methods of encircling prey and employing bubble-net feeding tactics. This algorithm mirrors the phases of prey encirclement and exploration, incorporating these natural strategies into a computational framework for solving optimization problems [36,37]. The mathematical model underlying the WOA is outlined in the following sections.

#### 4.2.1. Encircling Prey

The strategy of encircling prey is mathematically modeled on the humpback whale’s innate ability to locate and surround its target. This is mathematically represented as follows: (1)$$\vec{D}=\left|\vec{C}\cdot {\vec{X}}^{*}\left(t\right)-\vec{X}\left(t\right)\right|.$$
(2)$$\vec{X}\left(t+1\right)={\vec{X}}^{*}\left(t\right)-\vec{A}\cdot \vec{D}.$$

Here, *t* denotes the iteration index and $\vec{D}$ represents the distance vector between a whale and the prey, with $\vec{X^{*}}\left(t\right)$ indicating the position vector of the optimal solution (prey) at iteration *t* and $\vec{X}\left(t\right)$ being the position vector of a whale. The coefficients $\vec{A}$ and $\vec{C}$ play crucial roles in adjusting the movement direction and length of the step towards the prey.

The coefficients are defined as in Equation (Equation 3).
(3)$$\vec{A}=2\vec{a}\cdot {\vec{r}}_{1}-\vec{a},$$
(4)$$\vec{C}=2\cdot {\vec{r}}_{2}.$$

In the equations above, $\vec{r_{1}}$ and $\vec{r_{2}}$ are random vectors with values in the range [0, 1]. The vector $\vec{a}$ is a linearly decreasing coefficient from 2 to 0 as the iterations proceed, as defined by the following: (5)$$a=2-\frac{2t}{T_{\max}}.$$

This decreasing nature of $\vec{a}$ ensures that the algorithm shifts its focus from exploration to exploitation, effectively mimicking the transition from searching for prey to encircling and capturing it as the iteration count *t* approaches the maximum number of iterations $T_{\max}$.

Figure 4 illustrates the strategic maneuvering of a whale within a two-dimensional space, adapting its coordinates (X, Y) towards the optimal position (X*, Y*) identified within the same iteration. This adaptation is facilitated through the modulation of coefficients $\vec{A}$ and $\vec{C}$, which are influenced by the stochastic vectors ${\vec{r}}_{1}$ and ${\vec{r}}_{2}$. Consequently, the whale’s subsequent position in the iterative cycle is determined within a defined range, as depicted in Figure 4. At each iteration, the whale optimizes its spatial location, progressively converging towards the current optimum, thereby emulating the natural predatory behavior of encircling its prey. This principle is scalable to multidimensional spaces, where the whale navigates the hyperdimensional landscape to encircle the optimal solution point, with “n” denoting the dimensionality of the problem’s parameter space, thus ensuring a comprehensive exploration and exploitation in the search for the global optimum.

#### 4.2.2. Bubble-Net Attacking Method

In simulating the sophisticated hunting strategy of humpback whales, which includes a blend of encircling contraction and spiral movement towards the prey, mathematical formulations are developed to model both the shrinking encircling mechanism and the spiral approach strategy:

**1. Shrinking encircling mechanism:** This strategy is modeled by progressively diminishing the magnitude of the $\vec{a}$ parameter in Equation (Equation 3), where the value of $\vec{A}$ oscillates within the range [−a,a]. The decrement of *a* is linear as per Equation (Equation 5), facilitating a simulated contraction towards the prey. Specifically, when $\vec{a}$ lies within [−1,1], the subsequent position of the whale is determined within the bounds defined by its current location and the optimal target position. As depicted in Figure 5, within a two-dimensional framework, if $\vec{A}$ falls within [0,1], the prospective position of the whale is constrained to the region demarcated by points (X, Y) and (X*, Y*).

**2. Spiral position update:** As depicted in Figure 6, this method starts by calculating the gap between the whale’s present coordinates (X, Y) and the prime location (X*, Y*). A logarithmic spiral route is then established between these coordinates, mimicking the humpback whale’s helical approach to its prey. The principal formula for this helical trajectory is presented in Equation (Equation 6).
(6)$$\vec{X}\left(t+1\right)={\vec{D^{\prime }}}_{i}\cdot e^{bl}\cdot \cos\left(2\pi l\right)+{\vec{X}}^{*}\left(t\right),$$
where ${\vec{D^{\prime }}}_{i}={\vec{X}}^{*}\left(t\right)-\vec{X}\left(t\right)$ denotes the distance between the ith whale’s coordinates and the optimal position, *b* is a constant shaping the spiral’s contour, and *l* is a random variable within [−1,1].

To accurately reflect the dual nature of the humpback whale’s hunting behavior, incorporating both the spiral path and the constricting circle pattern, the model posits a 50% likelihood for each strategy during the optimization process. This balanced approach is essential for optimizing the whale’s position through iterations, as formalized below: (7)$$\vec{X}\left(t+1\right)=\left\{\begin{array}{cc} {\vec{X}}^{*}\left(t\right)-\vec{A}\cdot \vec{D}, & \mathrm{if}\;p<0.5\\ {\vec{D^{\prime }}}_{i}\cdot e^{bl}\cdot \cos\left(2\pi l\right)+{\vec{X}}^{*}\left(t\right), & \mathrm{if}\;p\geq 0.5 \end{array}\right.,$$
where *p* is a random number selected from the interval [0,1], facilitating the stochastic selection between the shrinking encirclement and the spiral approach.

#### 4.2.3. Search for Prey

The Whale Optimization Algorithm (WOA) employs a nuanced search strategy that includes both targeted and exploratory behaviors to effectively navigate the solution space. In this context, the algorithm incorporates a random search mechanism distinct from its bubble-net attacking strategy. During this phase, a whale’s position is updated by diverging from the location of a randomly chosen peer from the population rather than moving towards it. This approach is designed to widen the exploration area, fostering a more comprehensive search and mitigating the risk of premature convergence on suboptimal solutions. The mathematical representation of this behavior is given by the following: (8)$$\begin{array}{c} \vec{D}=\left|\vec{C}\cdot \vec{X_{\mathrm{rand}}}-\vec{X}\right|\\ \vec{X}\left(t+1\right)=\vec{X_{\mathrm{rand}}}-\vec{A}\cdot \vec{D} \end{array}.$$

Here, $\vec{D}$ denotes the distance vector between the current whale’s position and that of a randomly selected whale, indicated by $\vec{X_{rand}}$.

The Whale Optimization Algorithm (WOA) initializes with a diverse population of candidate solutions and iteratively refines these solutions through a simulated foraging behavior of humpback whales. The algorithm’s core mechanic, the parameter $\vec{A}$, dynamically decreases from 2 to 0 over the course of iterations. This decrement facilitates a shift from a detailed local search towards a more expansive global search as the algorithm progresses. The WOA employs two primary search strategies based on a stochastic parameter *p*: the shrinking encircling mechanism, which mimics the humpback whales’ bubble-net hunting method, and the Spiral Update path, emulating the helix-shaped movement of whales approaching prey. These strategies are probabilistically selected to update the positions of the “whales” within the search space, guiding them towards the optimum solution. The algorithm terminates when it meets specified end conditions, such as a maximum number of iterations or a satisfactory error threshold. The sequential steps of the WOA, delineated in the accompanying pseudocode, encapsulate the algorithm’s systematic approach to optimization, as outlined below (Algorithm 1).
**Algorithm 1** Whale Optimization Algorithm1:**Input:** The fitness of each individual whale2:**Output:** The position of the whale with the best fitness in the current iteration3:Initialize the whales population $X_{i}$ (i=1,2,...,40)4:The maximum number of iterations is set to 405:Calculate the fitness of each individual whale and retain the optimal individual6:$X^{*}=$ The position of the whale with the best fitness in the current iteration7:**while** (t< maximum number of iterations) **do**8:    For the position of each whale during the current iteration update parameters a,A,C,l and *p*9:    **if** p<0.5 **then**10:        **if** |A|<1 **then**11:           update position according to Equation (Equation 1)12:        **else**13:           select a random whale location $X_{\mathrm{rand}}$14:           update position according to Equation (Equation 8)15:        **end if**16:    **else**17:        update position according to Equation (Equation 6)18:    **end if**19:    Check if any whale individual exceeds the search space and amend it20:    Calculate the fitness of each individual whale and retain the optimal individual21:    Update $X^{*}$ if there is a better solution22:    t=t+123:**end while**24:**return** $X^{*}$

### 4.3. Whale Optimization Algorithm Improvement

The traditional Whale Optimization Algorithm (WOA) is known for its simplicity, minimal control parameters, and effectiveness in avoiding local optima, making it widely applicable in various domains. However, it has been observed that the WOA may suffer from limitations in terms of its convergence speed and solution accuracy. To address these concerns, an Adaptive Whale Optimization Algorithm incorporating Chaos Mapping, named the C-I-WOA-CNN model, has been introduced.

#### 4.3.1. Chaotic Mapping Whale Optimization Algorithm

Chaos mapping generates chaotic sequences through the iterative application of certain functions, exhibiting properties such as nonlinearity, a high sensitivity to initial conditions, strong ergodicity, and unpredictability. These mappings are often employed for population initialization to enhance the diversity and randomness of the algorithm, improving its overall performance [38,39].

In the evaluation of 16 chaotic optimization algorithms, the Cubic mapping was found to have a maximum Lyapunov exponent that exceeds those of the Sine, Circle, Liebovitch, Intermittency, Singer, and Kent mappings. It is comparable to the Logistic, Sinusoidal, Tent, Gauss, and Bernoulli mappings but is lower than the Chebyshev, ICMIC, and Piecewise mappings [40,41]. Despite this, the Cubic mapping was chosen for optimizing the traditional WOA due to its balance of performance and practicality, considering the complexity of its formula and the characteristics of the chaotic sequences it produces. The Cubic mapping is defined as in Equation (Equation 9).
(9)$$x_{n+1}=\rho x_{n}\left(1-x_{n}^{2}\right).$$

Here, ρ denotes the control parameter. This enhancement aims to leverage the dynamic and unpredictable nature of chaotic sequences to potentially improve the WOA’s convergence speed and the accuracy of its solutions.

#### 4.3.2. Combining Multiple Strategies in Whale Optimization

In the framework of the Whale Optimization Algorithm, achieving a balance between exploration (searching for prey) and exploitation (bubble-net attacking strategy) is crucial for enhancing the algorithm’s performance in terms of its convergence speed and solution precision. The dynamic interplay between these strategies, governed by the algorithm’s parameters, dictates its ability to efficiently navigate the search space and avoid premature convergence [35].

The incorporation of an adaptive inertia weight mechanism, inspired by its successful application in various meta-heuristic algorithms, offers a method to modulate the algorithm’s search behavior over time [42]. This mechanism adjusts the algorithm’s tendency to explore new areas or exploit known regions based on the progress of the search process. The adaptive inertia weight is mathematically represented as follows: (10)$$\omega =(\omega_{max}-\omega_{min})*mm*e^{(-\tau /t_{max})}+\omega_{min}.$$
where ω is the inertia weight, $\omega_{max}$ and $\omega_{min}$ are the maximum and minimum inertia weights, respectively, iter is the current iteration, and ${\mathrm{iter}}_{\max}$ is the maximum number of iterations. This adaptive approach allows for a more flexible and efficient search process, initially favoring global exploration and gradually shifting towards local exploitation as the search progresses.

By integrating this adaptive mechanism, the enhanced Whale Optimization Algorithm can dynamically adjust its behavior, effectively balancing between the diverse strategies of encircling prey and bubble-net attacking. The improved algorithmic steps, incorporating the adaptive inertia weight, are defined as in Equation (Equation 11).
(11)$$\vec{X}\left(t+1\right)=\left\{\begin{array}{cc} \omega {\vec{X}}^{*}\left(t\right)-\vec{A}\cdot \vec{D}, & \mathrm{if}\;p<0.5\;\mathrm{and}\;|\vec{A}|<1,\\ \omega {\vec{D}}^{\prime }\cdot e^{bl}\cdot \cos\left(2\pi l\right)+{\vec{X}}^{*}\left(t\right), & \mathrm{if}\;p\geq 0.5,\\ \omega {\vec{X}}_{\mathrm{rand}}-\vec{A}\cdot \vec{D}, & \mathrm{if}\;p<0.5\;\mathrm{and}\;|\vec{A}|\geq 1, \end{array}\right.$$
where *p* is a random number in [0, 1], $\vec{A}$ and $\vec{D}$ are coefficient vectors, ${\vec{X}}^{*}$ is the position vector of the best solution found so far, and ${\vec{X}}_{\mathrm{rand}}$ is a position vector chosen randomly from the current population. This approach enhances the algorithm’s adaptability and efficiency, ensuring a more robust search process capable of achieving high-quality solutions.

## 5. Enhanced Temperature Error Mitigation

### 5.1. Experimental Setup for Pressure Measurement

In our experimental framework, we employed a Honeywell ASDX100D44R pressure sensor combined with PT100-RTD temperature sensors. The setup was meticulously calibrated to measure temperatures ranging from 0 °C to 78 °C at 6 °C intervals, resulting in 14 distinct temperature points. Pressure measurements spanned from 0.4 MPa to 0.8 MPa, at 0.05 MKPa intervals, yielding nine discrete pressure points. Through extensive calibration, we amassed over 2000 datasets. Table 1 showcases a subset of the calibration data for the pressure sensor, indicating the applied pressure (p), the sensor’s voltage output (Up), and the corresponding temperature measurement (Ut), providing a comprehensive overview of the sensor’s performance across a spectrum of conditions.

### 5.2. Temperature Compensation Approach

We designed a Convolutional Neural Network (CNN) that takes the voltage output (Up), the measured temperature (Ut), and the measured pressure (p) as inputs, predict the applied pressure (p). To optimize the network’s learning efficiency and accuracy, we strategically selected calibration data at a range of specific temperatures (0 °C, 6 °C, 18 °C, 24 °C, 36 °C, 42 °C, 54 °C, 60 °C, 72 °C, and 78 °C) for the training phase. Conversely, data collected at intermediate temperatures of 12 °C, 30 °C, 48 °C, and 66 °C were earmarked for the testing phase to validate the model’s performance under varied thermal conditions. All input data were normalized within a 0-1 scale to facilitate optimal network training conditions.

In the domain of Convolutional Neural Network (CNN) model optimization, the delineation of learning rate and batch size parameters emerges as pivotal. The learning rate orchestrates the model’s pace of acquisition through the training trajectory, while the batch size plays a crucial role in modulating the model’s fitting prowess and generalization aptitude. Hence, the calibration of these parameters is paramount. We employ the Whale Optimization Algorithm (WOA) to ascertain the optimal settings for the learning rate and batch size, with both the population size and the iteration threshold of the algorithm set to 40. This strategy is further refined through the adjustment of the Cubic mapping function’s control parameter and the adaptive weight adjustment coefficient to unity, intending to equilibrate the exploration and exploitation phases within the optimization continuum. Following optimization, the optimal learning rate was determined to be 0.0064, and the optimal batch size was established at 36.

During the CNN training process, our network architecture was structured to include two convolutional layers, with the first and second layers having four and eight channels, respectively. The ReLU was utilized as the activation function throughout, and a fully connected layer was integrated at the conclusion of the network. For optimization, we employed the Adam optimizer, selecting a learning rate of 0.0064 and a batch size of 36. Both parameters were finely tuned and optimized through the Chaotic-Initiated Whale Optimization Algorithm (C-I-WOA) method. Additionally, the dataset was split into a training set and a test set at a ratio of 3:1.

We conducted multiple training sessions, initially training for 30 epochs with the optimized learning rate. Subsequently, we manually adjusted the learning rate to 80% of its original value and trained for another 30 epochs. This process was repeated five times, ultimately yielding a stable model. Table 2 displays some of the pressure measurements at different temperatures after algorithmic correction.

### 5.3. Comparative Analysis of Optimized Neural Network Models

To validate the efficacy of the proposed algorithm, we compared five neural network models, CNN, WOA-CNN, C-WOA-CNN, I-WOA-CNN, and C-I-WOA-CNN, using the test data for training. Figure 7 and Figure 8 depict the comparison of predicted versus actual pressure values and the error analysis post-optimization, respectively.

The analysis indicates that the C-I-WOA-CNN model exhibits the lowest error margins, maintaining an error range within ±2.5. Table 3 summarizes the error metrics for different temperature compensation models, showcasing the superior performance of the C-I-WOA-CNN model across several evaluation criteria.

The analysis derived from Table 3 illustrates that the C-I-WOA-CNN temperature compensation model surpasses alternative approaches in terms of the R-Squared, RPD, mean square error, and mean absolute percentage error values, showcasing its superior predictive capabilities. The R-Squared value, a measure of the model’s ability to capture the variance in the dataset, approaches 1 for the C-I-WOA-CNN model, indicating a high level of accuracy in predictions. The Ratio of Performance to Deviation (RPD), which compares the standard deviation of the validation set to the standard error of prediction, further reinforces the model’s reliability. RPD values exceeding 3, as observed with the C-I-WOA-CNN model, denote a robust predictive model capable of making precise and reliable forecasts. This high RPD value underscores the model’s exceptional performance in temperature compensation, emphasizing its potential for accurate and dependable applications in pressure measurement and other related fields.

## 6. Conclusions

In this study, we introduced an advanced temperature compensation technique for downhole pressure measurements in slim-hole drilling, leveraging the innovative C-I-WOA-CNN framework. This framework synergizes the adaptability of Convolutional Neural Networks (CNNs) with the robust optimization prowess of the Whale Optimization Algorithm (WOA), further augmented by chaotic mapping and adaptive mechanisms. Our approach overcomes the traditional challenges faced by CNNs, such as the entrapment in local optima and protracted convergence times, by endowing the model with superior exploratory and exploitative strengths.

The C-I-WOA-CNN model stands out for its exceptional accuracy and resilience, showcasing a marked reduction in error rates alongside a heightened prediction reliability in the realm of downhole pressure measurement. Notably, the model’s integration of chaotic dynamics and adaptive weighting not only enriches the solution space but also fine-tunes the balance between global and local search strategies, culminating in a faster convergence and enhanced model robustness.

Our findings herald a significant step forward in the domain of drilling technology, offering a robust framework for temperature compensation that could be extended to other complex measurement challenges within the petroleum industry and beyond. The C-I-WOA-CNN model’s adaptability and precision pave the way for future explorations into other applications where temperature fluctuations pose a significant challenge to measurement accuracy.

## Figures and Tables

**Figure 1 sensors-24-02162-f001:**
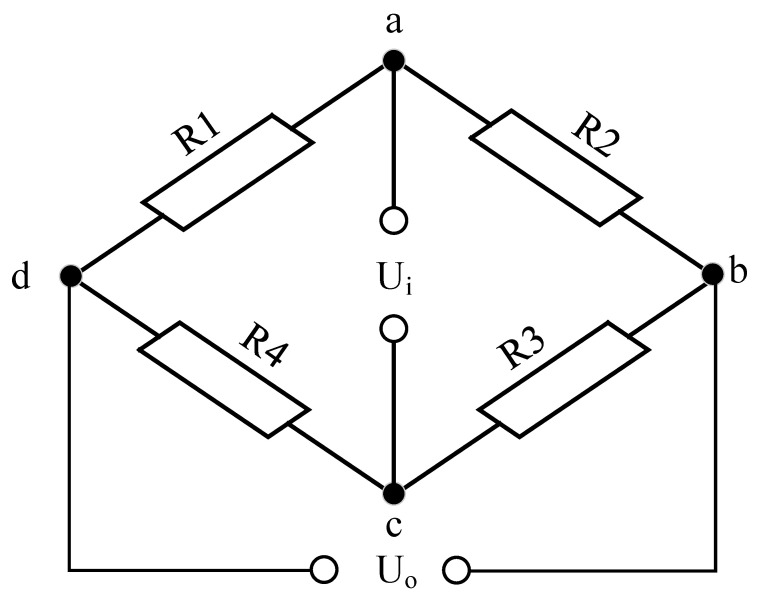
Schematic diagram of Wheatstone bridge.

**Figure 2 sensors-24-02162-f002:**
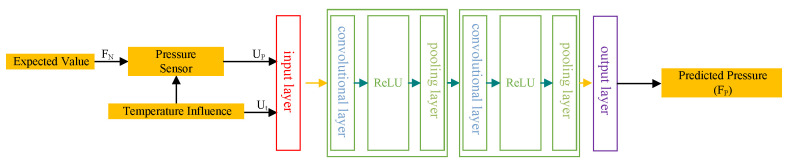
Structure of CNN for temperature compensation.

**Figure 3 sensors-24-02162-f003:**
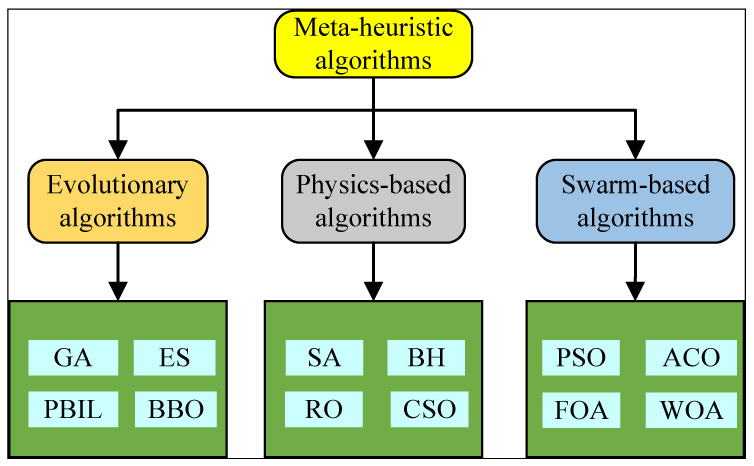
Types of meta-heuristic algorithms.

**Figure 4 sensors-24-02162-f004:**
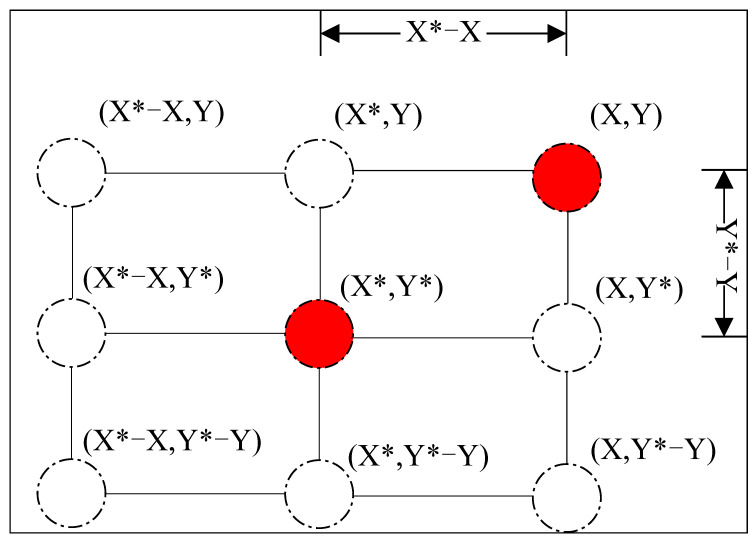
Schematic diagram of encircling prey. The red dots represent the positions in the current iteration cycle, with X, Y being the positions at this cycle, and X*, Y* being the optimal positions in the current iteration.

**Figure 5 sensors-24-02162-f005:**
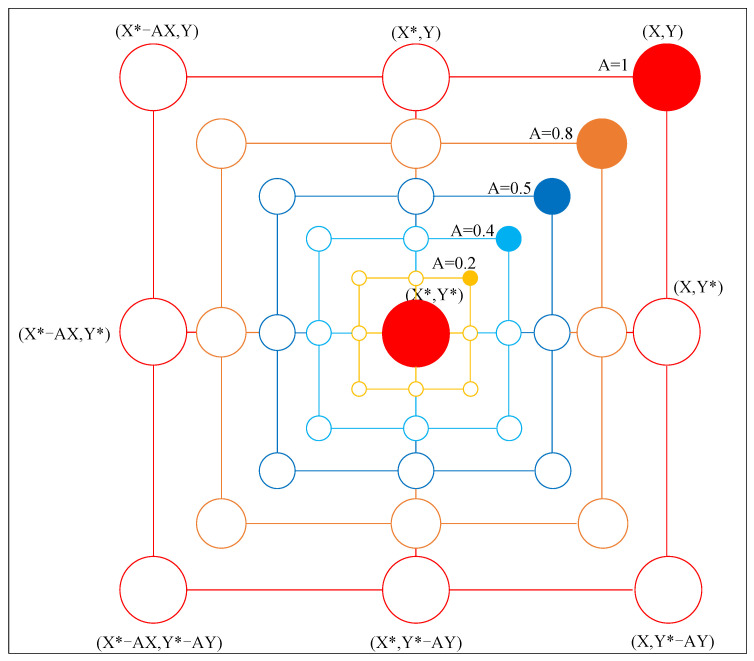
Visualization of the shrinking encircling mechanism in a two-dimensional space, the red dot (X, Y) represents the current position of the whale, (X*, Y*) represents the optimal position, and dots of other colors represent the position of the whale when the value of A varies.

**Figure 6 sensors-24-02162-f006:**
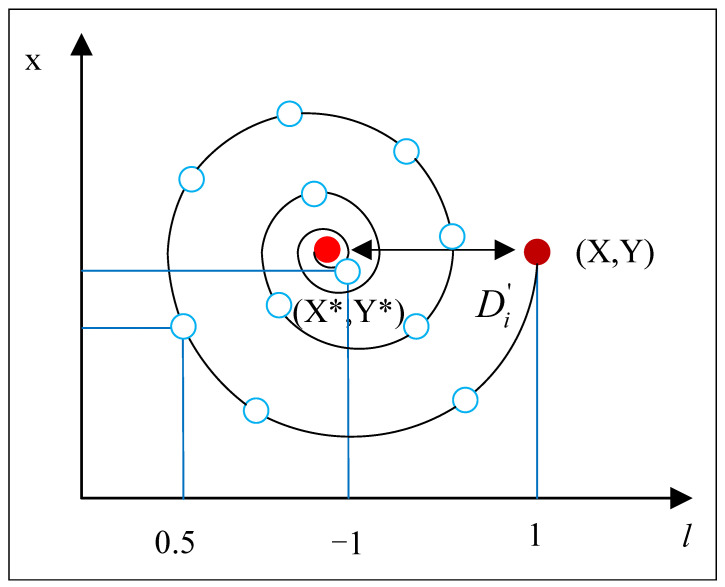
Spiral Updating Position strategy in action, the red dot (X, Y) represents the current position of the whale, (X*, Y*) represents the optimal position.

**Figure 7 sensors-24-02162-f007:**
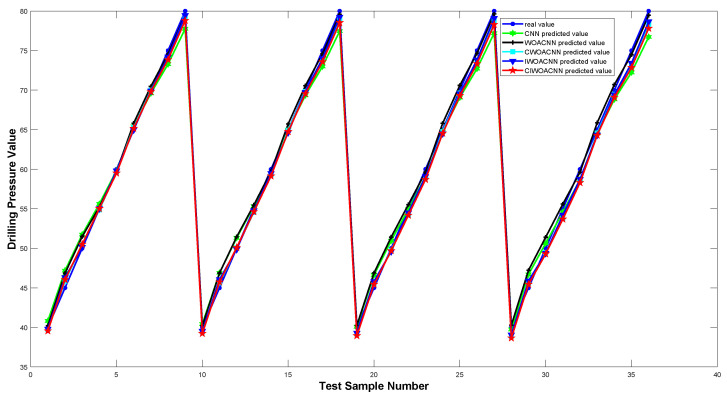
Predicted vs. Actual pressure values for optimized CNN models.

**Figure 8 sensors-24-02162-f008:**
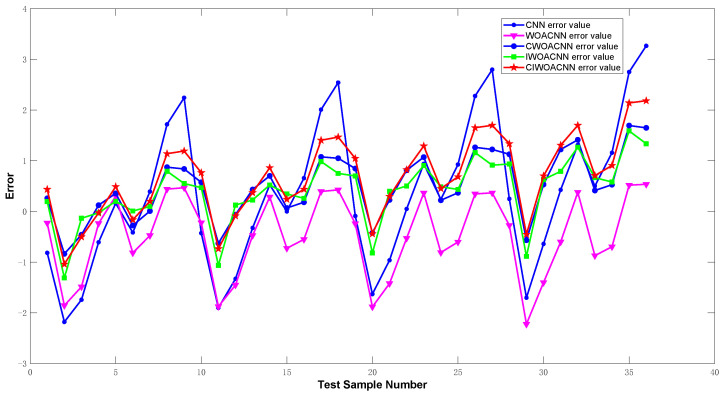
Error analysis for optimized CNN models.

**Table 1 sensors-24-02162-t001:** Pressure sensor calibration data.

p/MPa	0 °C		6 °C		12 °C		18 °C		24 °C		30 °C		36 °C	
	**Up/mV**	**Ut/°C**	**Up/mV**	**Ut/°C**	**Up/mV**	**Ut/°C**	**Up/mV**	**Ut/°C**	**Up/mV**	**Ut/°C**	**Up/mV**	**Ut/°C**	**Up/mV**	**Ut/°C**
0.4	83.24	0.34	83.41	6.59	86.95	12.45	87.123	18.57	88.56	24.51	90.46	30.21	91.36	36.32
0.45	95.71	0.37	96.37	6.61	98.36	12.51	101.54	18.62	103.18	24.56	104.84	30.41	106.12	36.41
0.5	104.8	0.42	106	6.65	108.03	12.56	111.32	18.65	112.24	24.62	113.87	30.43	115.54	36.52
0.55	113.4	0.46	115.6	6.73	118.37	12.58	120.51	18.67	122.56	24.75	123.67	30.47	125.17	36.56
0.6	124.4	0.57	129.1	6.76	130.09	12.62	131.86	18.71	133.23	24.78	134.51	30.62	135.62	36.61
0.65	136.5	0.63	138.5	6.79	141.39	12.65	143.23	18.73	144.04	24.83	146.13	30.64	147.86	36.65
0.7	147.9	0.74	149.4	6.83	151.31	12.68	152.64	18.78	153.62	24.87	155.05	30.71	156.63	36.68
0.75	155.8	0.78	157.3	6.87	158.55	12.71	160.46	18.81	162.87	24.76	164.05	30.75	165.74	36.71
0.8	164.8	0.81	167.7	6.93	168.31	12.73	172.04	18.83	173.16	24.51	174.83	30.83	176.04	36.76
**p/MPa**	**42 °C**		**48 °C**		**54 °C**		**60 °C**		**66 °C**		**72 °C**		**78 °C**	
	**Up/mV**	**Ut/°C**	**Up/mV**	**Ut/°C**	**Up/mV**	**Ut/°C**	**Up/mV**	**Ut/°C**	**Up/mV**	**Ut/°C**	**Up/mV**	**Ut/°C**	**Up/mV**	**Ut/°C**
0.4	92.65	42.65	94.14	48.36	95.65	54.36	96.04	60.24	96.74	66.45	97.45	72.24	98.24	78.46
0.45	107.8	42.68	109.2	48.39	110.36	54.39	110.87	60.26	111.63	66.47	112.31	72.26	113.79	78.48
0.5	117	42.71	118.5	48.42	119.65	54.42	121.24	60.34	121.87	66.51	122.47	72.29	123.14	78.51
0.55	126.7	42.73	128.2	48.46	129.41	54.48	130.63	60.39	131.42	66.56	131.89	72.34	132.41	78.53
0.6	137.1	42.76	138.8	48.51	139.87	54.51	140.74	60.42	141.36	66.59	141.76	72.37	142.32	78.57
0.65	149.2	42.82	150.6	48.56	151.64	54.53	152.68	60.47	153.13	66.62	153.89	72.41	154.65	78.62
0.7	158.1	42.86	159.6	48.63	161.13	54.56	161.97	60.51	162.75	66.63	163.45	72.46	164.27	78.65
0.75	167.2	42.87	168.7	48.68	169.65	54.57	170.12	60.58	170.84	66.69	171.53	72.48	171.82	78.68
0.8	176.5	42.93	178	48.74	179.21	54.63	180.04	60.64	180.76	66.84	181.46	72.53	181.76	78.74

**Table 2 sensors-24-02162-t002:** Compensated pressure measurements across varied temperatures using C-I-WOA-CNN model.

P/MPa	12 °C	30 °C	48 °C	66 °C
0.40	39.8323	39.6606	39.5244	39.3895
0.45	46.0485	45.9305	45.8047	46.0165
0.50	50.764	50.5898	50.431	50.2723
0.55	55.2128	55.2261	55.0571	54.8855
0.60	59.6937	59.5285	59.2875	59.0914
0.65	65.4543	65.2311	65.1976	65.1327
0.70	70.2645	70.2091	70.1342	70.0941
0.75	74.4586	74.373	74.2958	73.982
0.80	79.5653	79.473	79.4085	79.099

**Table 3 sensors-24-02162-t003:** Error Metrics for Different Temperature Compensation Models.

Model	R-Squared	RPD	Mean Square Error	Mean Absolute Percentage Error
CNN	0.98708	9.0489	1.1641	1.91%
WOA-CNN	0.99485	12.2413	0.74358	1.38%
C-WOA-CNN	0.99603	14.0599	0.67828	1.14%
I-WOA-CNN	0.99657	14.2482	0.64091	1.11%
C-I-WOA-CNN	0.99806	18.7464	0.48678	0.87%

## Data Availability

Data are contained within the article.

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
