# Peer review of "Advancing Slim-Hole Drilling Accuracy: A C-I-WOA-CNN Approach for Temperature-Compensated Pressure Measurements"

_sensors, 2024, doi:10.3390/s24072162_

Round 1

Reviewer 1 Report

Comments and Suggestions for Authors

Dear Dr.Fei Wang

The document titled "Advancing Slim-Hole Drilling Accuracy: A C-I-WOA-CNN Approach for Temperature-Compensated Pressure Measurements" introduces a novel approach to enhance the accuracy of downhole pressure measurements in slim-hole drilling operations within the petroleum industry. The paper proposes a temperature compensation model utilizing a Chaotic-Initiated Adaptive Whale Optimization Algorithm (C-I-WOA) to optimize Convolutional Neural Networks (CNN), termed the C-I-WOA-CNN model. The findings demonstrate that the C-I-WOA-CNN model outperforms traditional CNNs in terms of convergence speed, global search, and local exploitation, significantly reducing the average absolute percentage error in pressure parameter predictions. The paper highlights the importance of temperature compensation in drilling operations and presents the C-I-WOA-CNN model as a reliable solution to mitigate temperature-induced measurement inaccuracies.

Make the following suggestions for the paper:

Please provide a clearer explanation on how to train the CNN network and data set.

After the C-I-WOA-CNN model is improved, direct data that directly shows the effectiveness of the optimization is applied.

Comments on the Quality of English Language

The overall language expression is good, and the writing is standardized and logical.

Please pay attention to tense issues in grammar and problems with fluent sentences.

Author Response

Thank you very much for your review. The targeted revisions are in the replied document.

Reviewer 2 Report

Comments and Suggestions for Authors

In the process of drilling wells, and especially small diameter wells, an error in determining the bottomhole pressure can lead to fracturing of rocks. Hydraulic fracturing of rocks in the process of drilling leads to complications and accidents that can lead to drilling stoppage, oil and gas blowout and wellbore loss. Well pressure control under high bottomhole temperatures is important to prevent complications.

The solutions proposed in the paper to improve the accuracy of measurements are interesting and relevant. But in the process of familiarization I have some comments and suggestions:

1.       The paper does not describe the techno-technological scheme of sensor placement. It is not specified where the algorithms work directly, at the bottomhole or on the surface. What communication channel is planned to be used for data transmission on the surface and with what frequency (hydraulic, electromagnetic, wire or fiber optic?). Please consider these specifics.

2.       The actual pressure in the well reaches values two orders of magnitude higher than the pressures discussed in the article.

3.       Table 1 shows a small amount of data that may not be sufficient to train a neural network. No reference to the physical conditions of the process is given. Such a model may not work predictably or universally. The idea of implementing machine learning is good, but we need a dedicated and versatile algorithm.

4.       The learning algorithm should be based on real physical laws and experimental input data. In this case we get the calculated result corrected by statistical algorithms on the basis of real data.

5.       In Figure 7 and 8 the legend is not readable; the font size should be increased.

Author Response

(The authors gave the same response as above.)
